# Histone variant H2A.Z is needed for efficient transcription-coupled NER and genome integrity in UV challenged yeast cells

**Hélène Gaillard**[1,2]*, **Toni Ciudad**[3], **Andrés Aguilera**[1,2], **Ralf E. Wellinger**[1,2]

**1** Centro Andaluz de Biología Molecular y Medicina Regenerativa—CABIMER, Consejo Superior de Investigaciones Científicas—Universidad de Sevilla—Universidad Pablo de Olavide, Seville, Spain, **2** Departamento de Genética, Facultad de Biología, Universidad de Sevilla, Seville, Spain, **3** Departamento de Ciencias Biomédicas, Facultad de Ciencias, Universidad de Extremadura, Badajoz, Spain

* gaillard@us.es

## Abstract

The genome of living cells is constantly challenged by DNA lesions that interfere with cellular processes such as transcription and replication. A manifold of mechanisms act in concert to ensure adequate DNA repair, gene expression, and genome stability. Bulky DNA lesions, such as those induced by UV light or the DNA-damaging agent 4-nitroquinoline oxide, act as transcriptional and replicational roadblocks and thus represent a major threat to cell metabolism. When located on the transcribed strand of active genes, these lesions are handled by transcription-coupled nucleotide excision repair (TC-NER), a yet incompletely understood NER sub-pathway. Here, using a genetic screen in the yeast *Saccharomyces cerevisiae*, we identified histone variant H2A.Z as an important component to safeguard transcription and DNA integrity following UV irradiation. In the absence of H2A.Z, repair by TC-NER is severely impaired and RNA polymerase II clearance reduced, leading to an increase in double-strand breaks. Thus, H2A.Z is needed for proficient TC-NER and plays a major role in the maintenance of genome stability upon UV irradiation.

## Author summary

The genome of living organisms is constantly challenged by intrinsic and extrinsic DNA damaging agents. The resulting DNA lesions must be readily repaired to maintain genome integrity. This is particularly important for bulky DNA lesions, such as those produced by UV light, as they will block the progress of elongating RNA polymerases on transcribed genes. These DNA lesions are repaired by a specific pathway called transcription-coupled nucleotide excision repair (TC-NER), the dysfunction of which is associated with severe human diseases. In this work, we used budding yeast as a eukaryotic model organism to perform a genetic screen for new TC-NER factors. We discovered that the *HTZ1* gene, encoding the histone variant H2A.Z, is required for efficient DNA repair by TC-NER. Our molecular and genetic analyses showed that in the absence of H2A.Z, RNA polymerases persist on damaged DNA, causing interference with DNA replication and genome

**Data Availability Statement:** The authors confirm that all data underlying the findings are fully available without restriction. All relevant data are

within the paper and its Supporting Information files.

**Funding:** Research was funded by the Spanish Ministry of Science, Innovation and Universities (PID2022-140466NB-I00 to HG and RW), the Junta de Andalucía (P20_01220 to RW), the University of Seville (PP2018-10767 and PP2019-13299 to HG), the Spanish Ministry of Science and Innovation (BFU2016-75058-P to AA), the Spanish Ministry of Economy and Competitiveness (BFU2013-42918-P to AA), and the European Regional Development Fund (FEDER). The funders had no role in study design, data collection and analysis, decision to publish, or preparation of the manuscript.

**Competing interests:** The authors have declared that no competing interests exist.

instability. Our findings about the contribution of histone variant H2A.Z to the repair of UV damage further highlight the importance of appropriate chromatin environment for the maintenance of genome integrity.

## Introduction

Living cells continuously suffer DNA damage that may lead to mutations and genomic instability if left unrepaired. DNA damage may occur by exogenous agents such as ionizing radiation, ultraviolet (UV) light or chemicals or by endogenous factors derived from the cell metabolism. DNA damage generates structural distortions that interfere with basic cellular functions like transcription and replication. Cells possess a number of pathways to keep DNA lesions, transcription and replication stress under control, many of which are highly conserved throughout evolution. Among the repair pathways, nucleotide excision repair (NER) is a versatile repair pathway capable of removing a large variety of structurally unrelated lesions such as UV-induced cyclobutane pyrimidine dimers (CPDs) and pyrimidone 6–4 photoproducts, X-ray induced cyclopurines or adducts induced by chemicals such as 4-nitroquinoline oxide (4-NQO), benzo[a]pyrene, N-acetoxy-2-actylaminofluorene, psoralens, etc. These bulky lesions, which lead to RNA polymerase (RNAP) stalling when located on the template strand, can affect gene expression and may have severe consequences for cell function [1]. Furthermore, trapped RNAPs represent an impediment for the replication machinery and can lead to transcription-replication conflicts (TRCs), a primary source of genome instability [2,3]. Hence, bulky DNA lesions exhibit strong mutagenic and thus tumorigenic potential, as exemplified by skin cancer, which is primarily caused by exposure to natural UV light [4].

Transcription-blocking lesions are mainly repaired by transcription-coupled NER (TC-NER), which differs from global genome NER (GG-NER) in the lesion recognition step, while the core repair reaction is common to both sub-pathways. In TC-NER, stalling of the elongating RNAP at the DNA damage promptly triggers the repair reaction on the transcribed strand (TS) of active genes while in GG-NER, a specialized DNA damage recognition complex–which consists of Rad7, Rad16 and Elc1 in budding yeast–improves the detection of the helix-distorting DNA lesions throughout the genome. In eukaryotes, the human Cockayne's syndrome protein B (CSB) and its yeast homolog Rad26 are among the first proteins to act at DNA damage-stalled RNAPII and contribute to the recruitment of further repair factors [5,6]. Residual TC-NER activity persists in the absence of Rad26 in yeast and has been shown to depend largely on Rpb9, a nonessential subunit of RNAPII [7]. A handful of factors involved in different aspects of transcription, including elongation, formation of export-competent mRNA ribonucleoprotein complexes and termination are required for TC-NER proficiency [8–10]. Recent works have shown that the Elf1/ELOF1 transcription elongation factor functions in TC-NER both in yeast and human cells, where it promotes the recruitment and assembly of further repair factors [11–13]. Interestingly, these studies also revealed a role for ELOF1 in preventing transcription-replication conflicts upon treatment with genotoxic agents.

Despite the advances that have been made in our understanding of TC-NER, the actions taking place at a stalled RNAPII during TC-NER and the crosstalk with other nuclear events remain elusive [1,6]. With the aim at identifying new TC-NER factors to shed light on the molecular mechanisms of this repair pathway, we screened the non-essential deletion strain collection from *S. cerevisiae* for mutations leading to increased sensitivity to 4-NQO. This genetic screen was performed in strains lacking the *RAD7* GG-NER gene as a way to enhance the sensitivity of the assay and enrich in new functions involved in TC-NER. We identified

*HTZ1*, which encodes the yeast H2A.Z histone variant, as a gene required for efficient TC-NER and RNAPII clearance following UV irradiation. Importantly, UV-irradiation of H2A.Z-depleted cells leads to an accumulation of double-strand breaks (DSBs), which constitute a major threat to genome stability.

## Results

### Yeast KO collection screening for mutants impaired in TC-NER

With the aim at identifying factors with a yet undescribed role in TC-NER, we undertook a genome-wide screen for deletion mutants showing increased sensitivity to the UV-mimetic drug 4-NQO in a GG-NER defective strain background. Therefore, a yeast collection of 5140 viable knock-out strains [14] was crossed with a strain bearing a deletion of the *RAD7* repair gene and the growth of the resulting double mutants in control and 4-NQO containing plates assessed by visual inspection (Fig 1A). 820 mutants with apparent growth impairment when combined with *rad7Δ* were retrieved and screened a second time using the same conditions, leading to the identification of 44 candidate strains with putative roles in TC-NER (S1 Table). A selection of 31 strains were then assayed by manual cross with *rad7Δ*, tetrad dissection and analysis of UV and 4-NQO sensitivities, resulting in 19 candidates with confirmed phenotypes (Figs 1B and S1 and S1 Table). Among the identified genes, several have known function in DNA recombination, post-replicative repair, and DNA damage checkpoint, while others encode proteins without known implication in DNA repair-related processes. Next, we assessed the sensitivity of the corresponding single mutant strains to other genotoxic agents such as hydroxyurea (HU), camptothecin, caffein and methyl methanesulfonate (MMS) to gain insights into the possible functions of their gene product in different pathways (S1 Fig). Because DNA damage generated by the alkylating agent MMS is repaired by mechanisms other than NER, the 8 candidates that were not sensitive to MMS (*mph1Δ*, *hir1Δ*, *htz1Δ*, *clb5Δ*, *doc1Δ*, *fes1Δ*, *gim5Δ*, and *qri7Δ*) were selected for further analyses. The MMS sensitive *nup84Δ* strain, which was recently reported to be deficient both in TC- and GG-NER [15], was also included as NER-deficient control. The repair efficiency of UV-induced CPDs on the TS and non-transcribed strand (NTS) of the constitutively expressed *RPB2* gene were assessed (S2 Fig). A strain in which the *RPB9* gene is deleted was used as control, as this transcription factor is known to be required for efficient TC-NER [7]. All mutants showed repair efficiencies that were comparable to the wild-type strain, except NER-deficient *nup84Δ* and *htz1Δ*, for which repair on the TS was as low as for *rpb9Δ*.

The *HTZ1* gene encodes the yeast histone variant H2A.Z (also called Htz1), which has been implicated in numerous processes including DSB repair by homologous recombination (HR), replicative stress induced DSB formation and GG-NER in H2A.Z-bearing nucleosomes of an inactive promoter [16–20], but so far not in TC-NER. Because of the ubiquitous role of H2A.Z in DNA-templated processes, we generated double mutants lacking the HR factor Rad52, the post-replicative repair factor Rad18, or the endonuclease Rad1, which is strictly required for both NER sub-pathways, TC-NER and GG-NER. Sensitivity to 4-NQO and UV were assessed in these strains and in the corresponding double mutants lacking Rpb9, with the aim at evaluating possible overlap among repair pathways (Fig 1C). Deletion of *HTZ1* led to increased 4-NQO/UV sensitivity in cells deficient in HR, post-replicative repair, and NER, consistent with a role for H2A.Z in more than one single repair pathway. Noteworthy, similar results were obtained in cells deleted for Rpb9, suggesting that both gene products may have shared functions. However, no combination of these mutants could be assessed as *htz1Δ* and *rpb9Δ* are synthetic lethal [21](S2 Fig).

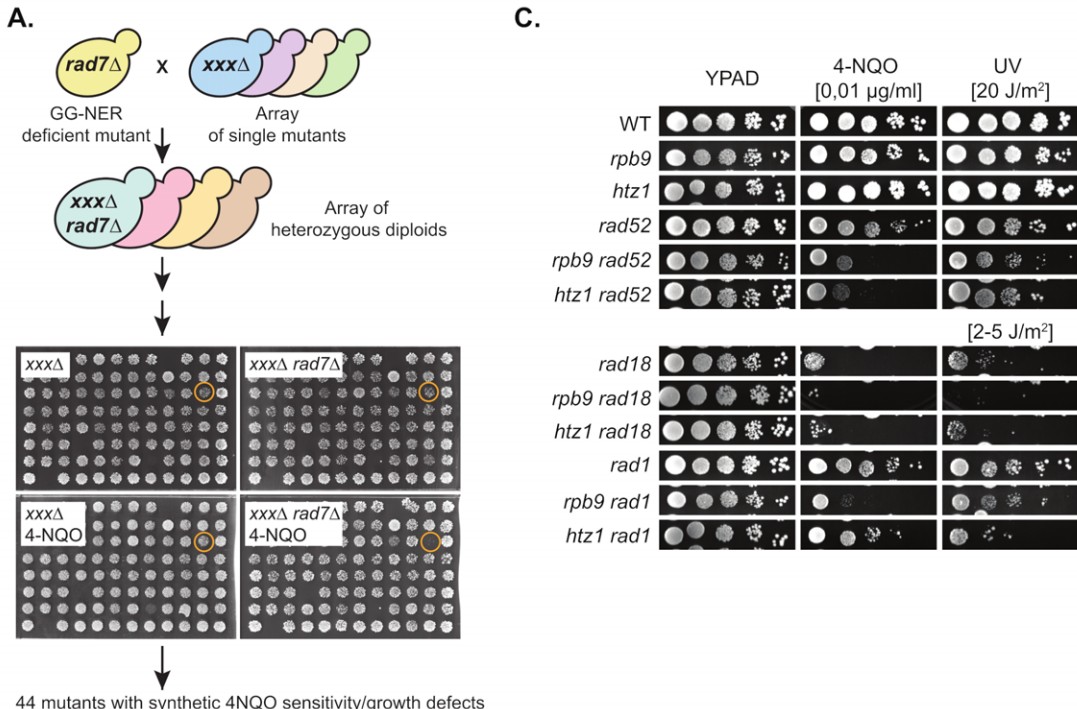

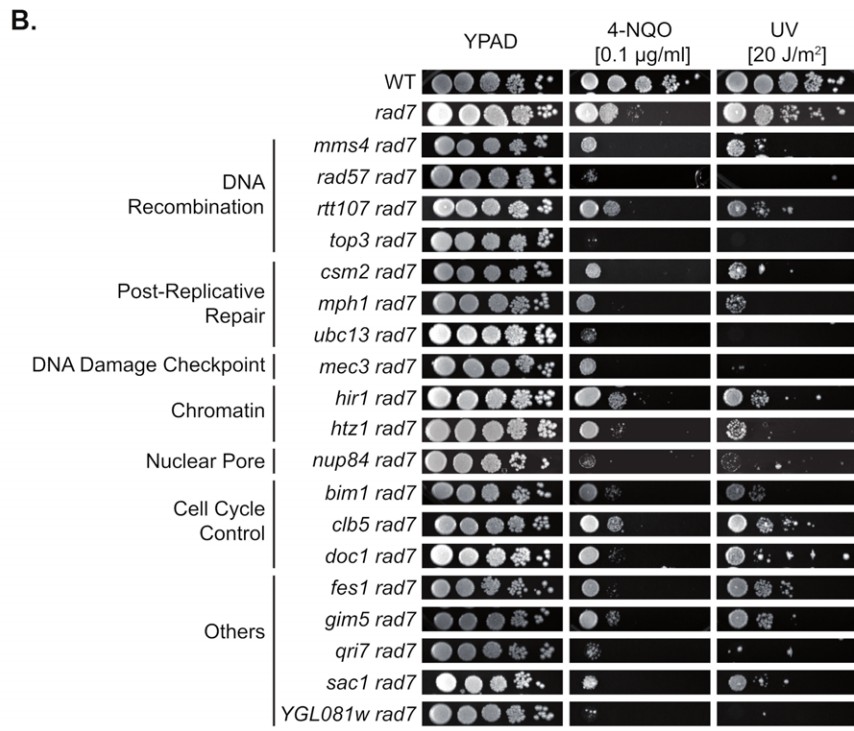

**Fig 1. Genetic screen for mutants sensitive to 4-nitroquinoline oxide in the absence of GG-NER factor Rad7. A.** Schematic overview of the synthetic lethal screen performed by crossing a *rad7Δ* strain with the yeast KO collection and selection of haploid spores on plates containing 0.01 μg/ml 4-nitroquinoline 1-oxide (4-NQO). Representative images of one of the plates are shown in which one strain showed synthetic 4-NQO sensitivity (orange circle). The 44 mutants identified are listed in S1 Table. **B.** Analysis of 4-NQO and UV sensitivity of 19 double mutants lacking the GG-NER factor Rad7. Serial dilutions of exponentially growing cultures were spotted on

YPAD plate supplemented or not with 0.1 μg/ml 4-NQO. Where indicated, plates were irradiated with 20 J/m$^2$ UV-C light. Wild-type (WT) and *rad7Δ* single mutants were used as control. Pictures were taken after incubating the plates in the dark for 3 days. Growth of the corresponding single mutants is shown in S1 Fig. **C.** Analysis of 4-NQO and UV sensitivity of single and double mutant combination of *rpb9Δ* and *htz1Δ* with DNA repair mutants *rad52Δ*, *rad18Δ* and *rad1Δ*. 4-NQO concentration and UV doses are indicated. Other details as in (B).

## Histone variant H2A.Z is required for efficient TC-NER

To gain further insights into the genetic interaction between H2A.Z and the GG-NER factor Rad7, sensitivity to 4-NQO, UV, MMS and menadione was assessed in single and double *htz1Δ rad7Δ* mutants (Fig 2A). The double mutant was more sensitive to each of these DNA damaging drugs than the respective single mutants. While the increased sensitivity to 4-NQO and UV confirms the results of our genetic screen (see Fig 1B), the high sensitivity to MMS and menadione suggests an additional role of the GG-NER factor Rad7 in the repair of damage other than bulky adducts in the absence of H2A.Z. Next, we analyzed the repair of UV-induced CPDs in the TS and the NTS of a 4.4-kb restriction fragment containing the constitutively expressed *RPB2* gene in wild-type and *htz1Δ* cells (Fig 2B). Our results demonstrate that TS repair is severely impaired in the absence of the H2A.Z histone variant, while NTS repair appears only slightly affected, indicating that H2A.Z is required for TC-NER and facilitates GG-NER at transcribed genes. Analysis of *RPB2* mRNA levels indicated that this gene is a little more expressed in *htz1Δ* than in wild-type cells (S3 Fig), excluding that transcriptional defects might account for the observed repair deficiencies.

Previous genome-wide studies of Htz1 occupancy showed that the constitutively expressed *RPB2* gene is not enriched in H2A.Z-bearing nucleosomes [22–24]. Therefore, we used a strain that expresses a Myc-tagged version of H2A.Z and a HA-tagged version of H2B to assess by chromatin immunoprecipitation (ChIP) whether H2A.Z deposition occurred in response to UV-induced damage (S3 Fig). During recovery from UV irradiation, the amount of H2A.Z *versus* H2B increased at the *RPB2* locus, in agreement with a role for this histone variant in the response to UV damage at transcribed genes. Since H2A.Z deposition is strictly dependent on the SWR1 chromatin remodeling complex in yeast [25], and that the absence of H2A.Z leads to faulty activity of this complex [26], we analyzed the impact of *SWR1* deletion on 4-NQO and UV sensitivity in *rad7Δ* and *htz1Δ* strains (S3 Fig). Importantly, equivalent growth inhibition upon exposure to 4-NQO or UV irradiation were observed in *htz1Δ rad7Δ*, *swr1Δ rad7Δ*, and *htz1Δ swr1Δ rad7Δ* cells, further supporting the idea that H2A.Z deposition is required for TC-NER proficiency.

Normal response to UV irradiation on transcribed genes encompasses RNAPII removal and transcriptional shut down, followed by DNA repair and resumption of transcription. To assess whether this response is affected in cells deficient in H2A.Z, we measured RNAPII binding to the *RPB2* gene upon UV irradiation by ChIP with antibodies raised against the RNAPII Rpb3 subunit (Figs 2C and S3). The results obtained with the wild-type reflect a proficient UV response involving a weak increase in RNAPII occupancy at the 5'-UTR and a drop within the gene body immediately upon irradiation followed by recovery of the transcriptional activity at timepoints 60 and 90 minutes. In *htz1Δ* cells however, RNAPII accumulated at the 5'-UTR and resumption of transcription was impaired. Altogether, these results indicate that H2A.Z is an important contributor to proficient TC-NER and that RNAPII may remain trapped on the DNA upon UV irradiation in the absence of this H2A variant.

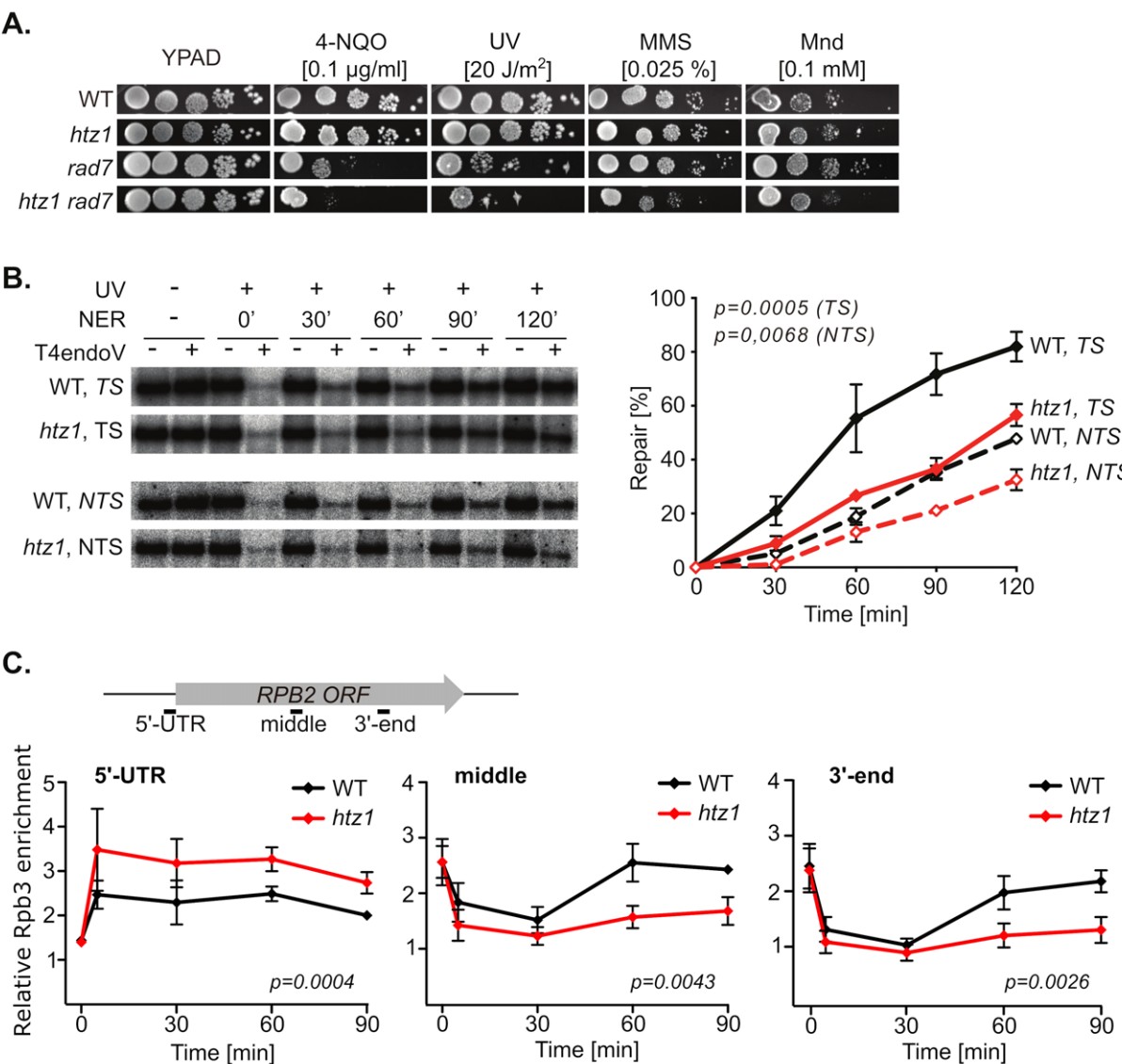

**Fig 2. Defective TC-NER and promoter-proximal accumulation of RNA polymerases in UV irradiated *htz1Δ* cells. A.** Growth analysis of wild-type (WT), *htz1Δ*, *rad7Δ* and *htz1Δ rad7Δ* strains upon exposure to different genotoxic agents. Serial dilutions of exponentially growing cultures were spotted on YPAD plate supplemented or not with 0.1 µg/ml 4-nitroquinoline 1-oxide (4-NQO), 0,025% methyl methanesulfonate (MMS), 0.1 mM menadione (Mnd). Where indicated, plates were irradiated with 20 J/m² UV-C light. Pictures were taken after incubating the plates in the dark for 3 days. **B**. Representative results of Southern analysis showing the repair of a 4.4-kb (*NsiI/PvuI*) *RPB2* fragment in WT and *htz1Δ* cells on the transcribed strand (*TS*) and the non-transcribed strand (*NTS*) (*left*). Non-irradiated DNA and DNA not treated with T4endoV were used as controls. Graphical representation of the quantified results is shown (*right*). Average values and standard error of the mean (SEM) derived from independent experiments are plotted (n = 3). The initial damage generated was on average 0.54 CPD/kb in the TS and 0.48 CPD/kb in the NTS. p-values (Wilcoxon, two-tailed) for repair on the TS and on the NTS are indicated. **C.** Binding of RNA polymerase II was analyzed by ChIP with an anti-Rpb3 antibody at the 5'-UTR, middle and 3'-end of the *RPB2* gene in WT and *htz1Δ* cells at the indicated timepoints after UV irradiation. A scheme of the amplicons is shown (*top*). Relative enrichment as compared to a non-transcribed region of chromosome V were calculated using the $2^{-\Delta\Delta Ct}$ method. Average values and SEM derived from independent experiments are plotted (n = 3). Statistical analyses were performed using a two-tailed Wilcoxon test. The amount of DNA obtained by ChIP with mouse IgG or Rpb3 antibody is shown in S3 Fig.

## H2A.Z plays a major role in the maintenance of genome integrity in UV irradiated cells

Previous studies have linked the absence of H2A.Z to increased genome instability [16,17,19,20,26]. Our findings that this histone variant is required for efficient TC-NER led us to ask whether UV irradiation may magnify genome instability in *htz1Δ* cells. Therefore, we used a plasmid-borne reporter system based on two truncated *leu2* inverted repeats to measure recombination frequency in control and UV-irradiated cells (Fig 3A). Cells lacking H2A.Z showed a moderate increase in recombination as compared to the wild-type, which was exacerbated upon UV irradiation (from 3-fold to 4.5-fold). Because conflicts between the transcription and replication processes have emerged as a major source of genome instability [2,3] and our results suggest that RNAPII might remain trapped on DNA in UV-irradiated *htz1Δ* cells, we decided to analyze recombination in a reporter system based on two truncated *leu2* direct repeats positioned either in a co-directional or head-on orientation with respect to replication (Fig 3B). In the absence of UV irradiation, similar recombination frequencies were obtained in *htz1Δ* and wild-type cells, yet the head-on orientation led to a recombination increase of about 3-fold as compared to the co-directional orientation regardless of the genetic background. Interestingly, UV-irradiation induced a significant increase in recombination which was much more pronounced in *htz1Δ* than in wild-type cells. This phenomenon was particularly evident with the head-on orientation recombination system (10-fold increase in *htz1Δ* as compared to 3-fold increase in WT), indicating that the absence of histone variant H2A.Z triggers genome instability upon UV irradiation that is aggravated by frontal encounters with the replication machinery.

Next, we asked whether *htz1Δ* might show genetic interaction with *rad26Δ* or *elf1Δ*, both of which are defective in TC-NER [5,12]. Deletion of *RAD26*, the yeast homologue of human *CSB*, increased the sensitivity of *htz1Δ* cells to 4-NQO and HU (Fig 4A). However, deletion of *ELF1*, the yeast homologue of human *ELOF1*, did not show an increased sensitivity to either of these genotoxic agents as compared to the respective single mutants. These results suggest that H2A.Z and Rad26 act in alternative pathways required to cope with DNA damage and/or during replication stress and that H2A.Z may function in Elf1-mediated pathways. As expected for GG-NER proficient cells, weak UV sensitivity was observed in all mutant strains. Survival analyses were therefore performed in GG-NER deficient cells (Fig 4B), revealing that *HTZ1* depletion slightly enhances the UV sensitivity of both *rad26Δ rad7Δ* and *elf1Δ rad7Δ* cells, indicating that the role of H2A.Z in TC-NER is not restricted to a specific sub-pathway.

To gain insight into the individual impact that the absence of H2A.Z, Rad26 and Elf1 may have on genome stability, we monitored the incidence of DSB repair centers by fluorescence microscopy upon UV irradiation. Cells were transformed with a plasmid expressing a YFP-tagged version of the recombination protein Rad52 and the percentage of S/G2 cells with one, two, and three or more Rad52-YFP foci at different timepoints following UV irradiation quantified in wild-type, *htz1Δ*, *rad26Δ* and *elf1Δ* cells (Fig 4C). In wild-type cells, the percentage of cells with DSB repair centers rose up to over 40% two hours after UV irradiation and went down to less than 30% at later timepoints, indicating that a substantial amount of DSBs is formed upon UV irradiation but that repair proficient cells recover readily. As expected for a strain with TC-NER defects, *rad26Δ* cells accumulated Rad52-YFP foci as the wild-type but showed a significant delay in recovering. Similar results were obtained with the *elf1Δ* strain, in agreement with previously described TC-NER defects associated with this mutation [12]. Notably, UV irradiated *htz1Δ* cells accumulated further DSB repair centers on top of their already elevated endogenous level and retained a high amount of cells with Rad52-YFP foci up to 6 hours after UV irradiation. Together, our results indicate that histone variant H2A.Z is required for TC-NER proficiency and that it plays a major role in the maintenance of genome stability in UV-irradiated cells.

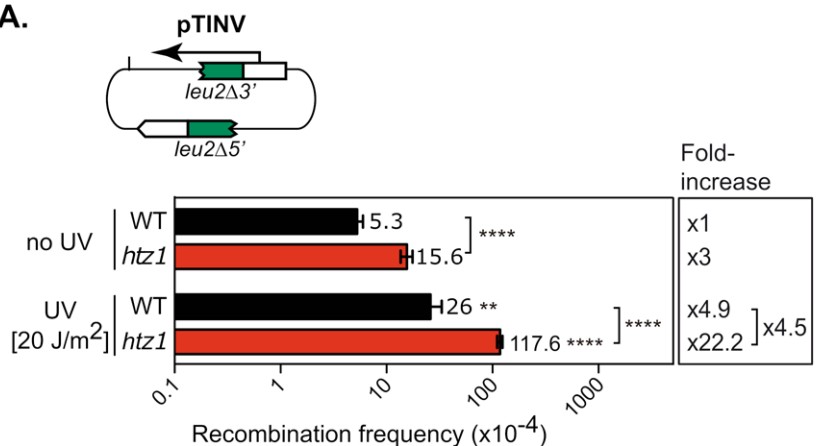

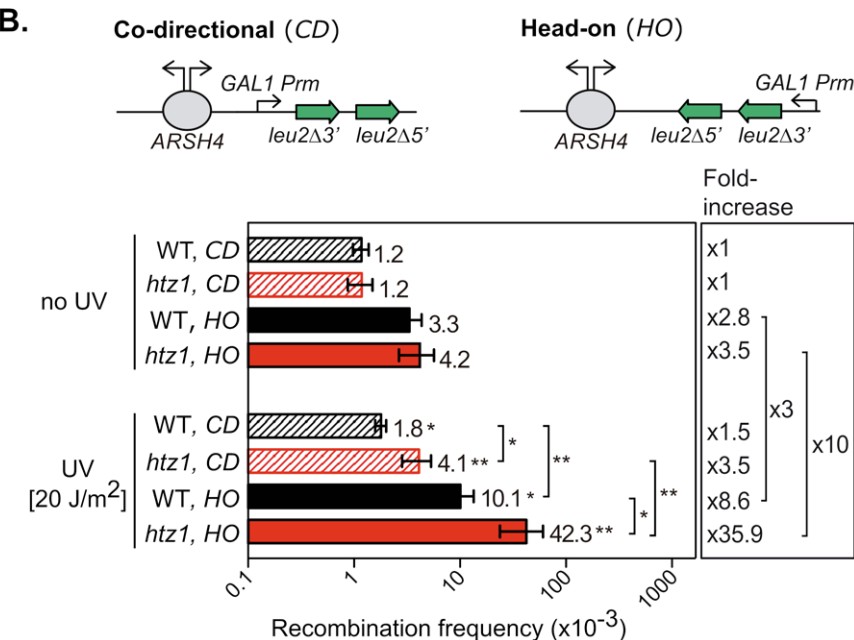

**Fig 3. Increased recombination frequencies in UV-irradiated *htz1Δ* cells. A**. Recombination analysis using the pTINV inverted-repeat plasmid-borne system in wild-type (WT) and *htz1Δ* cells. A scheme of the recombination system is shown (*top*). Where indicated, cells were irradiated with 20 J/m² UV-C light. Recombination frequencies were obtained as the median value of six independent colonies. The average and standard error of the mean of independent fluctuation tests are shown for each genotype (n≥3). Fold increases, as compared to the wild type without UV irradiation are shown (*left*). Statistical analyses were performed with a two-tailed unpaired student *t* test. * *P*<0.05, ** *P*<0.01; ***P*<0.001; ****P*<0.0001. **B**. Recombination analysis in WT and *htz1Δ* cells using a plasmid-borne direct-repeat recombination system oriented either in co-directional (*CD*) or in head-on (*HO*) orientation relative to the replication origin. Schemes of the recombination systems are shown (*top*). Other details as described in (A).

## Discussion

Using an unbiased classical genetic screening approach, we have identified histone variant H2A.Z as a new player in TC-NER. Our results support a role for this histone variant both in the regulation of RNAPII occupancy and TC-NER repair efficiency upon UV irradiation.

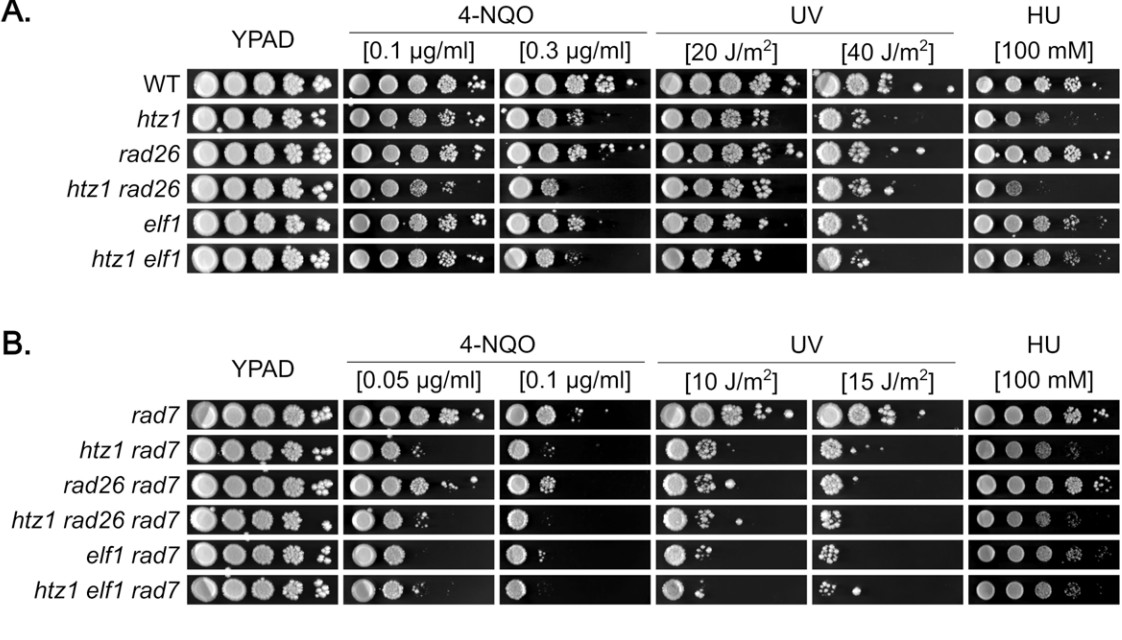

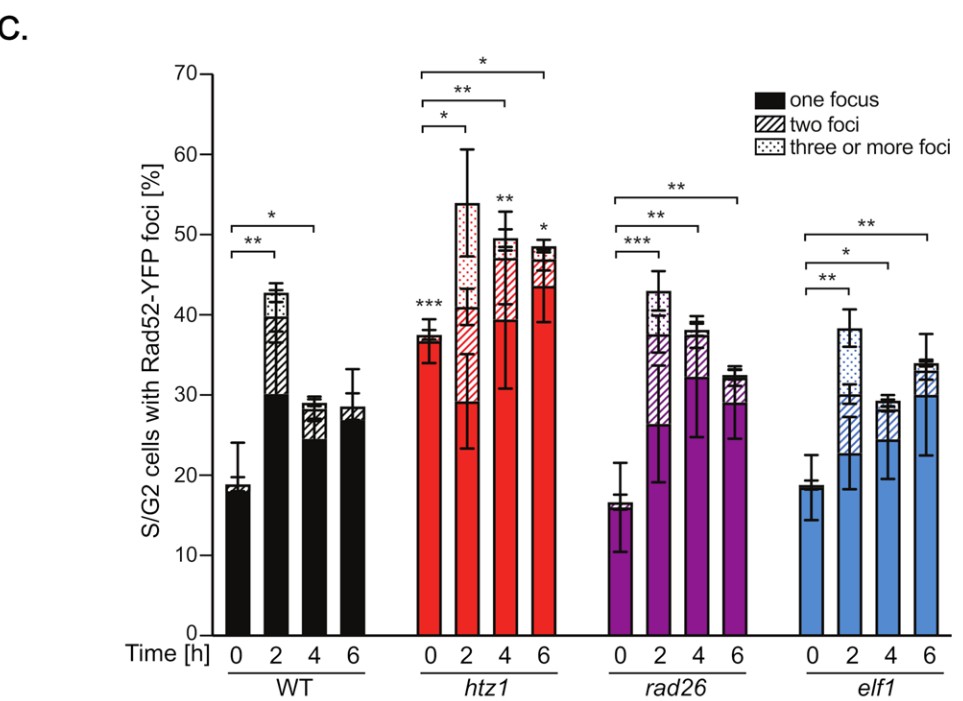

**Fig 4. *htz1Δ* is not epistatic to *rad26Δ* or *elf1Δ* and shows sustained Rad52 repair foci upon UV irradiation. A.** Sensitivity analysis to 4-nitroquinoline 1-oxide (4-NQO), UV and hydroxyurea (HU) in single and double mutant combination of *htz1Δ* with *rad26Δ* or *elf1Δ*. Serial dilutions of exponentially growing cultures were spotted on YPAD plate supplemented or not with the indicated concentration of 4-NQO or HU. Where indicated, plates were irradiated with UV-C light. Pictures were taken after incubating the plates in the dark for 3 days. **B.** Sensitivity analysis to 4-NQO, UV and HU in GG-NER defective cells (*rad7Δ*). Other details as in (A). **C.** Percentage of S/G2 cells containing Rad52-YFP foci in WT, *htz1Δ*, *rad26Δ* and *elf1Δ* cells and at the indicated timepoint after irradiation with 50 J/m² UV-C light. Average percentage of S/G2 cells with one Rad52-YFP focus, two Rad52-YFP foci and more than three Rad52-YFP foci obtained from independent transformants, and corresponding SEM are shown (n≥3). Statistical analyses were performed on the total % of cells with Rad52-YFP foci using an unpaired two-tailed *t*-test. * $P<0.05$, ** $P<0.01$; ***$P<0.001$.

Importantly, defective response to UV damage leads to sustained DSBs and genome instability in *htz1Δ* cells. H2A.Z is the only histone variant that is conserved from yeast to human and, while it is essential for proper mammalian development, it is not essential in budding yeast except in conditions requiring rapid transcriptional activation [27]. Previous studies have linked H2A.Z to the DNA damage checkpoint [28,29], DSB repair [26,30,31], chromosome segregation [32], DSB-free resolution of stalled replication forks [20] and GG-NER in H2A.Z-bearing nucleosomes at a repressed promoter [18]. While the participation of H2A.Z in DSB repair seems to be conserved in higher eukaryotes [33], the mechanisms underlying the function of this histone variant in the maintenance of genome stability remain largely unresolved.

The H2A.Z histone variant had not been linked to TC-NER so far, and to our knowledge only one report relates H2A.Z-containing nucleosomes located at an inactive promoter with increased GG-NER efficiency [18]. Increased repair was observed at the repressed *MFA2* promoter and relied on improved Gcn5-mediated histone H3 acetylation and recruitment of the Rad14 NER factor. Notably, this function was not observed at the silent *HMRa1* locus were H2A.Z is normally not present, suggesting that pre-existing H2A.Z-containing nucleosomes favor repair at repressed loci. Our CPD repair analysis on the constitutively active *RPB2* gene, which is not enriched in H2A.Z nucleosomes [22–24], revealed a weak repair delay on the NTS and a strong repair defect on the TS in *htz1Δ* cells. These defects likely arise because of a lack of H2A.Z incorporation by the SWR1 complex in response to UV damage, as Htz1 relative enrichment increases upon UV irradiation throughout *RPB2* and that *swr1Δ*, *htz1Δ* and *swr1Δ htz1Δ* cells are equally sensitive to UV in a GG-NER deficient background. The SWR1 remodeling complex has been shown to bind nucleosome-free DNA and the adjoining nucleosome core particle, enabling discrimination of gene promoters over gene bodies [34,35]. Stalling of elongating RNAPII at DNA damage sites may lead to nucleosome-free regions that could represent binding sites for SWR1, leading to the incorporation of H2A.Z into surrounding nucleosomes. H2A.Z-enriched chromatin, which enhances DNA accessibility [36], could facilitate repair either by increasing the surface available for interaction with NER proteins or by promoting the action of specific factors that would fuel repair. H2A.Z incorporation may also improve the DNA damage signaling or destabilize RNAPII binding by promoting post-translational modifications on histones or other proteins. Indeed, local changes in chromatin environment triggered by H2A.Z incorporation around a stalled RNAPII likely include Gcn5-mediated H3 acetylation, as reported at the repressed *MFA2* promoter [18]. This could account for the observed contribution of H2A.Z to NTS repair, as GG-NER on NTS of active genes has been shown to depend on Rad7-Rad16-mediated Gcn5 recruitment and H3 acetylation [37].

Our ChIP analysis indicates that RNAPIIs remain bound at the 5'-UTR region but not in the *RPB2* gene body in UV irradiated *htz1Δ* cells. This data can reflect either that RNAPIIs accumulate at the 5'-UTR due to defective TC-NER and recovery of transcription elongation, or that UV-dependent degradation of promoter-bound RNAPIIs–a critical pathway to avoid DNA-damage induced transcription stress [38]–does not take place correctly in the absence of histone variant H2A.Z. The latter would imply a direct role for this histone variant in the recruitment of the factors involved in UV-induced RNAPII degradation. Even though further investigation would be required to discriminate between these possibilities, the persistence of chromatin-bound RNAPIIs is predicted to lead to increased TRCs which may challenge genome stability. We found that recombination is greatly increased in a reporter system located in head-on orientation with respect to replication and sustained Rad52-foci are observed in UV-irradiated *htz1Δ* cells. These results are consistent with the idea that TC-NER defects and persistent stalled RNAPII represent a major barrier to replication fork progression. H2A.Z incorporation by Swr1 was shown to be important for the maintenance of DNA

integrity during replication stress [20]. We thus propose a model in which H2A.Z incorporation in the vicinity of RNAPII stalled at DNA lesions promotes efficient TC-NER repair by establishing a repair-favorable chromatin environment and, thus, preventing the accumulation of chromatin-bound RNAPII that could cause conflicts with the replication machinery in UV challenged cells (S4 Fig).

Worthy of note, other TC-NER factors were shown to function both in promoting efficient repair on transcribed genes and in avoiding harmful collisions between the replication and transcription machineries. For example, ELOF1 (Elf1 in yeast) not only promotes TC-NER but also protects cells against transcription-mediated replication stress upon DNA damage [12,13]. Another example is provided by the chromatin-reorganizing FACT (facilitates chromatin transcription) complex, which is required for the resolution of TRCs that are mediated by R-loops [39] and for proficient TC-NER in human cells [40]. Thus, TC-NER emerges as a process that may interfere with replication progression. Our finding that histone variant H2A.Z is required for efficient TC-NER and contributes to the prevention of UV-induced DSBs highlights the significance of DNA repair-optimized chromatin environment in maintaining genomic stability.

## Material and Methods

### Yeast strains, plasmids and growth conditions

All yeast strains and plasmids used in this study are listed in S2 and S3 Tables, respectively. Deletion mutants were obtained from the yeast deletion collection (Euroscarf) or obtained either by genetic crosses or by standard PCR-based gene replacement. For strains yHG122-1, yHG122-2, yHG122-11, gene replacement was achieved using a *rad1Δ*::*LEU2*-containing amplicon obtained from strain W839-6B (R. Rothstein). For strains yHG223-4, yHG224-3, yHG225-3 and yHG226-4, gene replacement was achieved using a *rad26Δ*::*HIS3*-containing amplicon obtained from strain MGSC102 [41]. For strain yHG240-3, gene replacement was achieved in yHG231-3C using a *rad7Δ*::*URA3*-containing amplicon obtained from strain yHG72-1B. Yeast cells were grown to mid-log phase in rich medium (YPAD; 1% yeast extract, 2% peptone, 2% glucose, 0.004% adenine sulfate) or synthetic defined medium (SD; 0.17% yeast nitrogen base, 0.5% ammonium sulfate, 2% glucose, 0.2% drop-out mix) at 30˚C.

### 4-Nitroquinoline oxide synthetic genetic array screen

We conducted our 4-NQO synthetic genetic array screen in 96-well plates. *MAT***a** haploids from the yeast KO collection bearing the G418 resistance marker [14](YSC1053, Open Biosystems) were mated in 100 μl liquid YPD with strain yHG84-2A and grown for 3 days at 30˚C without shaking. Diploid cells were selected in 1 ml SD-Trp-Ura medium. After 3 days incubation at 30˚C, cells were centrifuged, washed with sterile $H_2O$, resuspended in 600 μl sporulation media (1% potassium acetate supplemented with amino acids) and incubated for 6 days at 30˚ with vigorous shaking. Cells were then centrifuged, washed with sterile $H_2O$ and digested with zymolyase 20T (100 μg/ml) overnight at 30˚C with gentle shaking. Spores were then centrifuged, washed once and resuspended in 500 μl sterile $H_2O$. Using an automated robot (Hamilton Microlab Star robotics), spores were spotted on SD-Leu and SD-Leu-Ura plates supplemented with 600 μg/ml G418 to select for haploid cells bearing the KO deletion and the *rad7Δ* mutation, respectively. Plates were supplemented or not with 0.01 μg/ml 4-NQO. Growth was monitored upon incubation at 30˚C for 3 days. The entire procedure was repeated for 820 strains which showed an increased sensitivity to 4-NQO when combined with *rad7Δ*. The 44 final candidates are described in S1 Table.

## Drug sensitivity assays

Yeast cells grown to mid-log phase in YPAD were adjusted to an initial $A_{600}$ of 0.5, serially diluted 1:10, and spotted onto plates without or with the indicated genotoxic agents at the indicated concentrations. For UV irradiation, plates were irradiated in a BS03 UV irradiation chamber (Dr. Gröbel UV-Elektronik GmbH) at the indicated doses and incubated in the dark. Images were taken after 3 to 4 days growth at 30˚C. Two or more biological replicates were performed for all conditions.

## Gene- and strand-specific repair assays

CPD repair at the *RPB2* gene was analyzed as described [42]. Briefly, cells were grown at 30˚C in SD medium, irradiated in SD medium lacking amino acids with 150 J/m$^2$ UV-C light (BS03 UV irradiation chamber; Dr. Gröbel UV-Elektronik GmbH), the medium supplemented with amino acids and cells incubated at 30˚C in the dark for recovery. Isolated DNA samples were digested with *Pvu*I and *Nsi*I (Roche) and aliquots mock-treated or treated with T4-endonuclease V (T4endoV, Epicentre). DNA was electrophoresed in 1.3% alkaline agarose gels, blotted to Nylon membranes and hybridized with radioactively labelled strand-specific DNA probes, which were obtained by primer extension. Sequences of the primers are listed in S4 Table. Membranes were analyzed and quantified with a PhosphorImager (Fujifilm FLA5100). The remaining intact restriction fragment after treatment with T4endoV corresponds to the fraction of undamaged DNA. The CPD content was calculated using the Poisson expression, $-\ln(RF_a/RF_b)$, where $RF_a$ and $RF_b$ represent the intact restriction fragment signal intensities of the T4endoV- and mock-treated DNA, respectively. Repair curves were calculated as the fraction of CPDs removed versus time. The initial damage was set to 0% repair.

## Chromatin immunoprecipitation

Chromatin immunoprecipitation was performed as described [42]. Briefly, cells were grown at 30˚C in SD medium, irradiated in SD medium w/o amino acids with 150 J/m$^2$ UV-C light (BS03 UV irradiation chamber; Dr. Gröbel UV-Elektronik GmbH), the medium supplemented with amino acids and the cells incubated at 30˚C in the dark for recovery. Cells were broken in a multibead shocker at 4˚C in lysis buffer (50 mM HEPES-KOH pH 7.5, 140 mM NaCl, 1 mM EDTA pH8, 1% Triton X-100, 0.1% sodium deoxycholate) supplemented with 1x Complete Protease Inhibitor Cocktail (Roche) and 1 mM PMSF. Chromatin was sonicated to an average fragment size of 400–500 bp in a Bioruptor (Diagenode). Samples were centrifuged to eliminate cell debris. An aliquot was processed as Input and the chromatin immunoprecipitated overnight at 4˚C with anti-Rpb3 RNAPII subunit (1Y26, Neoclone), anti-HA (ab9110, Abcam) or anti-Myc (9E10, Takara Bio) antibodies coated to Protein A or G Dynabeads (Invitrogen). Mouse IgG (SIGMA) coated Dynabeads were used as control. Real-time quantitative PCR was performed using iTaq universal SYBR Green (Biorad) with a 7500 Real-Time PCR machine (Applied Biosystems). Standard curves for all pairs of primers were performed for each analysis. All PCR reactions were performed in triplicate. Relative Rpb3 enrichment was calculated using the formula $2^{-\Delta\Delta Ct} = 2^{-((Ct\,INPUT\,target - Ct\,IP\,target) - (Ct\,INPUT\,control - Ct\,IP\,control))}$. A non-coding region of chromosome V was used as control. Htz1 relative enrichment was calculated by normalization of the Myc-Htz1 IP signal ($2^{-\Delta Ct}$) with the HA-H2B IP signal ($2^{-\Delta Ct}$) obtained in parallel from the same chromatin extract. The primer sequences are listed in S4 Table.

## RT-qPCR analysis

Cells were grown to log phase at 30°C in SD medium and total RNA purified with RNAeasy kit (Qiagen) according to manufacturer instructions. 700 ng RNA were treated with ezDNase (Invitrogen) and converted to cDNA using SuperScript III Reverse Transcriptase (Invitrogen). Real-time quantitative PCR was performed using iTaq universal SYBR Green (Biorad) with a 7500 Real-Time PCR machine (Applied Biosystems). *SCR1* mRNA was used as internal control for comparative Ct calculation. Standard curves for all pairs of primers were performed for each analysis. All PCR reactions were performed in triplicate. The primer sequences are listed in S4 Table.

## Recombination assays

Recombination frequencies were determined as the average value of the median frequencies obtained from at least three independent fluctuation tests with the indicated recombination systems. Each fluctuation test was performed from six independent colonies according to standard procedures [43]. For UV-induced recombination, plates were irradiated with 20 J/m$^2$ UV-C light (BS03 UV irradiation chamber; Dr. Gröbel UV-Elektronik GmbH) prior to incubation.

## Detection of Rad52-YFP foci

Rad52-YFP foci were visualized in cells transformed with plasmid pWJ1213 with a DM600B microscope (Leica) as previously described [44] with minor modifications. Individual transformants were grown to early-log-phase in SD-Leu, irradiated in SD medium lacking amino acids with 50 J/m$^2$ UV-C light (BS03 UV irradiation chamber; Dr. Gröbel UV-Elektronik GmbH), the medium supplemented with amino acids and cells incubated at 30°C in the dark for recovery. Samples were fixed for 10 minutes in 0.1 M K$_i$PO$_4$ pH 6.4 containing 2.5% formaldehyde, washed twice in 0.1 M K$_i$PO$_4$ pH 6.6, and resuspended in 0.1 M K$_i$PO$_4$ pH7.4. At least 60 S/G2 cells were analyzed in each sample. Average values obtained from at least 3 independent transformants are plotted for each condition.

## Statistical analyses

Statistical analyses were performed using GraphPad Prism 7.0. Statistical significance was determined from at least 3 independent biological replicates using either Student's t-test or Wilcoxon signed-rank test. Two-tailed unpaired Student's t-test was used for comparison of the means of two different experimental conditions. Two-tailed Wilcoxon signed-rank test was used for the analyses of time course data. Differences with a P-value lower than 0.05 were considered significant. * p<0.05; ** p<0.01; *** p<0.001; **** p<0.0001. The number of independent experiments (n), specific statistical tests and significance are described in the Figure legends.

## Supporting information

**S1 Fig. Sensitivity of 19 selected mutants to various genotoxic agent.** Growth analysis of wild-type (WT) and 19 selected single mutants upon exposure to different genotoxic agents. Serial dilutions of exponentially growing cultures were spotted on YPAD plate supplemented or not with 0.1 μg/ml 4-nitroquinoline 1-oxide (4-NQO), 200 mM hydroxyurea (HU), 20 μg/ml camptothecin (CPT), 10 mM caffein, or 0,025% methyl methanesulfonate (MMS). Where indicated, plates were irradiated with 20 J/m$^2$ UV-C light. Pictures were taken after incubating the plates in the dark for 3 days. Strains that were selected for subsequent CPD repair analysis are indicated with an asterisk.
(PDF)

**S2 Fig. Short CPD repair kinetics to assess NER efficiency in 8 selected mutants. A**. Southern analysis showing the repair of a 4.4-kb (*Nsi*I/*Pvu*I) *RPB2* fragment in WT and 8 selected mutants on the transcribed strand (*TS*) and the non-transcribed strand (*NTS*). NER-defective *nup84Δ* and TC-NER defective *rpb9Δ* strains were used as controls. Non-irradiated DNA and DNA not treated with T4endoV were used as controls. **B.** Quantification of the CPD repair results shown in (A) is plotted for the TS (*left*) and NTS (*right*) (n = 1). **C.** Tetrad dissection analysis from a *htz1Δ rpb9Δ* diploid strain.
(PDF)

**S3 Fig. H2A.Z incorporation analyses. A**. *RPB2* expression analysis in WT and *htz1Δ* as determined by RT-qPCR at the *RPB2* middle locus. Average values and SEM are plotted (n = 3). Statistical analyses were performed with a two-tailed unpaired student *t* test. * $P < 0.05$ **B.** Analysis of relative H2A.Z enrichment at the 5'-UTR, middle and 3'-end of the *RPB2* gene upon UV irradiation. ChIP was performed in a strain bearing tagged versions of Htz1 and H2B. Relative enrichment was obtained by normalization of the immunoprecipitated Htz1 signal with the H2B signal obtained in parallel with the same extract. Non-coding chromosome V region (Chr. V) was used as control. Average values and SEM are plotted for each region (n = 2). Mouse IgG was used as negative control. **C.** Growth analysis of wild-type (WT) and single, double or triple mutants' combination of *htz1Δ*, *swr1Δ* and *rad7Δ* upon exposure to 4-nitroquinoline 1-oxide (4-NQO) and UV light. Serial dilutions of exponentially growing cultures were spotted on YPAD plate supplemented or not with 0.1 μg/ml 4-NQO. Where indicated, plates were irradiated with 20 J/m$^2$ UV-C light. Pictures were taken after incubating the plates in the dark for 3 days. **D.** Percentage of input DNA obtained by ChIP with mouse IgG or Rpb3 antibody in WT and *htz1Δ* cells at the indicated *RPB2* regions. Average values and SEM are plotted for each condition (n = 2).
(PDF)

**S4 Fig. *Model of H2A.Z function in TC-NER and at replication forks.*** Schematic drawing illustrating the function of H2A.Z-enriched chromatin in the vicinity of RNAPII stalled at a bulky DNA lesion and at replication forks in wild-type (WT) and *htz1Δ* cells. The RNAPII is drawn in yellow, the replisome schematized as green helicase and H2A.Z-containing nucleosomes highlighted in red.
(PDF)

**S1 Table. List of candidate genes.**
(XLSX)

**S2 Table. Strains used in this study.**
(PDF)

**S3 Table. Plasmids used in this study.**
(PDF)

**S4 Table. Primers used in this study.**
(PDF)

**S1 Information. Numerical row data.**
(XLSX)

## Acknowledgments

We thank A. Nicolas, B. Guillemette and M. Lisby for strains and plasmids.

## Author Contributions

**Conceptualization:** Hélène Gaillard, Andrés Aguilera, Ralf E. Wellinger.

**Data curation:** Hélène Gaillard.

**Formal analysis:** Hélène Gaillard.

**Funding acquisition:** Hélène Gaillard, Andrés Aguilera, Ralf E. Wellinger.

**Investigation:** Hélène Gaillard, Toni Ciudad.

**Project administration:** Hélène Gaillard.

**Supervision:** Hélène Gaillard, Andrés Aguilera.

**Visualization:** Hélène Gaillard.

**Writing – original draft:** Hélène Gaillard.

**Writing – review & editing:** Hélène Gaillard, Toni Ciudad, Andrés Aguilera, Ralf E. Wellinger.

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
