## [Decision Letter · Decision Letter 0]

11 Jun 2024

Dear Dr Gaillard,

Thank you very much for submitting your Research Article entitled 'The histone variant H2A.Z is required for efficient transcription-coupled NER and to maintain genome integrity in UV challenged yeast cells.' to PLOS Genetics.

The manuscript was fully evaluated at the editorial level and by independent peer reviewers. The reviewers appreciated the attention to an important problem, but raised some substantial concerns about the current manuscript. Most of them are about data presentation and discussion supporting significant advancement in understanding repair mechanisms. There are also suggestions about experimental validation and about data detailing.  Based on the reviews, we will not be able to accept this version of the manuscript, but we would be willing to review a much-revised version. We cannot, of course, promise publication at that time.

If you decide to revise the manuscript for further consideration at PLOS Genetics, please aim to resubmit within the next 60 days, unless it will take extra time to address the concerns of the reviewers, in which case we would appreciate an expected resubmission date by email to plosgenetics@plos.org.

We are sorry that we cannot be more positive about your manuscript at this stage. Please do not hesitate to contact us if you have any concerns or questions.

Yours sincerely,

Dmitry A. Gordenin, Ph.D.

Academic Editor

PLOS Genetics

Aleksandra Trifunovic

Section Editor

PLOS Genetics

Reviewer's Responses to Questions

**Comments to the Authors:**

Reviewer #1: The manuscript by Gaillard et al. reports the identification of the histone variant H2A.Z as a component in the resistance of yeast cells to 4-NQO and UV. The authors further demonstrate that H2A.Z plays a significant role in repairing UV lesions in the transcribed strand and, to a lesser extent, in the non-transcribed strand of the RPB2 gene, indicating a role for H2A.Z in TC-NER and GG-NER. The elimination of H2A.Z is epistatic to the deletion of Swr1, consistent with a previous study (doi: 10.1093/nar/gkt688) showing that Swr1 facilitates NER by recruiting H2A.Z to chromatin. These results support the idea that Swr1, along with H2A.Z, facilitates TC-NER and GG-NER by affecting their common substrates—lesions in chromatin. In line with its role in NER, H2A.Z appears to constrain recombination associated with UV damage, especially in situations where the transcription machinery collides with the DNA replication machinery.

The research was well planned and conducted, and the conclusions are largely well-supported by the experimental data. However, the findings presented are not novel. As noted by the authors, H2A.Z has been well-known to be implicated in numerous processes, including its role in NER. Therefore, I do not believe the findings are significant enough for publication in PLoS Genetics.

Reviewer #2: The manuscript by Gaillard et al screens >5K individual mutants to identify and characterize the involvement, and requirement of the gene encoding a highly conserved H2A.Z variant histone for efficient transcription-coupled NER. This is confirmed by well-designed plating assays and strand-specific DNA repair assays, and genomic instability assayed with reporter constructs. This is a well laid out study overall that ascribes a role of this histone variant selectively in TC-NER. The findings should be of interest fairly broadly. That said, there are some issues that should be addressed before the manuscript is ready.

Moderate issues.

Lines 110-113. What is the rationale or use in classifying mutants by categories? This seems arbitrary, as written,

Do authors see or expect to see changes in mRNA production of the target genes in the absence of H2A.Z? Could at least some defects in TC-NER be explained by changes in transcription?

Lines 137-141. An alternative explanation is that GG-NER can be also involved in the same repair in addition to TC-NER components. This should be considered when making conclusions from the data. This could potentially explain lack of complete epistasis as shown in the next section, for example.

I was not able to find information on, in authors’ opinion, how H2A.Z specifically contributes to TC-NER. Is this by recruiting some partners, changing nucleosome biophysical properties, enforcing Pol II termination, etc. What authors think happens should be specified in Discussion more clearly. For example, Line 268 states the fact, but offers no discussion as to why.

Perhaps the more substantial data presentation issue has to do with ChIP experiments, which are currently shown as either arbitrary units or ratios between antibodies, neither of which is informative. ChIP data should be presented as percent of input for each individual antibody, with ratios presented optionally. A no-antibody control should also be included. This applies to figure S3A and main 2C.

Minor points.

Are transcription-coupled NER and genome integrity roles of H2A.Z coming from the same mechanism or are two independent roles? The title seems to suggest the latter, but it is not clear whether this distinction is made or really should be made based on the presented data.

Is it necessary to spend a good portion of Introduction on Elf1, which is barely mentioned in Results?

Fig 1A is described as showing a portion of 820 mutants with growth impairment when combined with tad7Delta. This is not seen in the actual Fig 1A example.

Fig 2A. It appears that rad7 individual mutant grows better than WT in the presence of MMS and Mnd. Do authors have an explanation and/or a more representative picture if it is an artifact?

Fig S1. Are these single mutants? Might be useful to highlight this in the legend. Also, nup84 mutant mentioned in the main text should probably be highlighted on the figure.

Line 44. What data suggest “chromatin plasticity” conclusion and what this means?

Line 51. Delete “severely” or delete the entire part of the sentence starting with “and thus”.

Line 73. Delete “trigger”.

Line 203. Delete “Interestingly”. The same can be done in most other instances throughout.

Line 227. Delete “apparently”.

Line 262. “Unstabilize”?

Some language should be proofread throughout, for example:

Line 50. “DNA damages generate” -> “DNA damage generates”

Lines 69-71. “In GG-NER” should be moved to after “while” to read: “while in GG-NER, a specialized DNA damage recognition complex…”

Line 104. “baring” should be “bearing”

Line 109. “in” should be “into”

Line 117. “encode for” should be either “encode” or “code for”

Line 181. “liked” should be “linked”

Line 191. “positioned”

Line 209. Delete “a”

Line 216. Delete “thus”.

Line 220. “raised” should be “rose”

Line 221. “Substantial”

Line 222. Delete “do”.

Line 258. Should be “DNA damage sites”

Line 259. “region” should be “regions”.

Line 260. “in” should be “into”.

Line 262. “eventhough” should be “even though”.

Reviewer #3: The manuscript by Gaillard et al describes a very nice initial characterization of a novel role for the H2A.Z histone variant in TC-NER. The authors conducted a systematic screen for increased 4NQO sensitivity in rad7 background, and found a number of interesting hits. They focused the subsequential analyses on arguably the most interesting of them all, HTZ1 encoding the H2A.Z homolog in S. cerevisiae. This particular histone variant has been shown previously to play multiple roles in genome stability, but never before in TC-NER. The work reported in this manuscript is going to be impactful because it uncovered yet another important facet of HA2.Z’s function in genome protection, so for this reason, I believe this work would receive a good amount of attention from a broad audience. The results of the screen itself (several additional genes) are also quite valuable as a resource that will be appreciated by the DNA repair community. The text is very nicely written and the data presented are quite clear and straightforward. I do not have any concerns about this work, and I only offer very minor suggestions for revisions to the text listed below.

Line 22: roadblocks (plural)

Line 55: structurally

Line 74: in (lower case i)

Line 104: bearing (not baring). “baring” is also incorrectly used as at other parts of the manuscript.

Line 155: It looks to me in Fig. 2B like the effect on the NTS is also pretty substantial. How would this look like in a GG-NER defective mutant? Perhaps add this point somewhere in the Discussion?

Line 176: “these results indicate that H2A.Z is required for proficient TC-NER”. I am not so sure about the use of “required” in this sentence. This is a subtle point, but I think this effect is better described as "contributes to" TC-NER, avoiding the term "required", which is potentially confusing to some readers. Consider changing to: “these results indicate that H2A.Z is an important contributor to proficient TC-NER”.

Line 181: “have liked the absence of H2A.Z with increased genome instability”; replace with have liked the absence of H2A.Z to increased genome instability”

Line 221: Substantial

Line 262: destabilize

Line 275: Swr1 (protein notation)

**Have all data underlying the figures and results presented in the manuscript been provided?**

Reviewer #1: Yes

Reviewer #2: **No: **Not clear if ChIP-qPCR raw data were provided

Reviewer #3: Yes

PLOS authors have the option to publish the peer review history of their article (what does this mean?). If published, this will include your full peer review and any attached files.

Reviewer #1: No

Reviewer #2: No

Reviewer #3: No

---

## [Decision Letter · Decision Letter 1]

26 Aug 2024

Dear Dr Gaillard,

We are pleased to inform you that your manuscript entitled "Histone variant H2A.Z is needed for efficient transcription-coupled NER and genome integrity in UV challenged yeast cells." has been editorially accepted for publication in PLOS Genetics. Congratulations!

Before your submission can be formally accepted and sent to production you will need to complete our formatting changes, which you will receive in a follow up email. Please, note though that Reviewer 2 suggested some small edits for typos.  We ask you to address those along with formatting changes.

Please be aware that it may take several days for you to receive this email; during this time no action is required by you. Please note: the accept date on your published article will reflect the date of this provisional acceptance, but your manuscript will not be scheduled for publication until the required changes have been made.

Yours sincerely,

Dmitry A. Gordenin, Ph.D.

Academic Editor

PLOS Genetics

Aleksandra Trifunovic

Section Editor

PLOS Genetics

Comments from the reviewers (if applicable):

Reviewer's Responses to Questions

**Comments to the Authors:**

Reviewer #1: In the revised manuscript, the authors have adequately addressed my major concerns regarding the novelty of their findings. While I do not find the manuscript particularly exciting, I now consider it acceptable for publication."

Reviewer #2: The manuscript is substantially improved after authors addressed the queries appropriately and in sufficient detail, and reads very well. I have a couple of small comments that do not need re-revision plus some more proofreading while I was reading it anyways.

Line 76-78 - the sentence is confusing: is dependence on Rpb9 attributed to Rad26?

Line 164. I think the word "only" should be kept.

Line 167. The word "discarding" is probably a wrong one

Line 188. I think it should be "consisting of" or better yet, "involving"

Lines 207-218. I may have missed this earlier, but Figure 3B numbers on the right are very confusing as presented. It is hard to follow the figure when reading the text referring to fold differences. Instead of vertical rectangles, especially the last two, could authors connect the relevant numbers with a line?

Line 280 - "allowing for"?

Line 284 "enhances accessibility to the DNA". The edits made this confusing as to accessibility of what to DNA.

Line 295 "Our ChIP analysis indicates"

Reviewer #3: The authors have adequately addressed all my comments and concerns.

**Have all data underlying the figures and results presented in the manuscript been provided?**

Reviewer #1: Yes

Reviewer #2: Yes

Reviewer #3: Yes

PLOS authors have the option to publish the peer review history of their article (what does this mean?). If published, this will include your full peer review and any attached files.

Reviewer #1: No

Reviewer #2: No

Reviewer #3: No

**Data Deposition**

http://datadryad.org/submit?journalID=pgenetics&manu=PGENETICS-D-24-00525R1

**Press Queries**

---

## [Editor Report · Acceptance letter]

4 Sep 2024

PGENETICS-D-24-00525R1 

Histone variant H2A.Z is needed for efficient transcription-coupled NER and genome integrity in UV challenged yeast cells. 

Dear Dr Gaillard, 

We are pleased to inform you that your manuscript entitled "Histone variant H2A.Z is needed for efficient transcription-coupled NER and genome integrity in UV challenged yeast cells." has been formally accepted for publication in PLOS Genetics! Your manuscript is now with our production department and you will be notified of the publication date in due course.

With kind regards,

Anita Estes

PLOS Genetics

On behalf of:
